# Incidence and Determinants of Caesarean Section in Shiraz, Iran

**DOI:** 10.3390/ijerph17165632

**Published:** 2020-08-05

**Authors:** Mahnaz Zarshenas, Yun Zhao, Colin W. Binns, Jane A. Scott

**Affiliations:** 1Fatemeh College of Nursing and Midwifery, Shiraz University of Medical Sciences, Shiraz 71348-14336, Iran; Mahnaz_zarshenas@yahoo.com; 2School of Public Health, Curtin University, Perth 6102, Australia; Y.zhao@curtin.edu.au (Y.Z.); C.Binns@curtin.edu.au (C.W.B.)

**Keywords:** caesarean section, elective, emergency, incidence, determinants

## Abstract

The rate of Caesarean section (CS) without medical indication has increased markedly worldwide in the past decades. This study reports the incidence of CS and identifies the determinants of elective and emergency CS as separate pregnancy outcomes in a cohort of Iranian women. Mothers (*n* = 700) of healthy, full-term infants were recruited from five maternity hospitals in Shiraz. The association between maternal socio-demographic and biomedical factors with mode of delivery was explored using multivariable, multinomial logistic regression. Most mothers underwent either an elective (35.4%) or emergency (34.7%) CS. After adjustment, women were more likely to deliver by elective CS than vaginally if they were older (≥30 year) compared to younger mothers (<25 year) (Relative Risk Ratio (RRR) 2.22; 95% Confidence Interval (CI) 1.28, 3.84), and had given birth at a private hospital (RRR 3.64; 95% CI 1.79, 7.38). Compared to those educated to primary or lower secondary level, university educated women were more likely to have undergone an elective (RRR 2.65; 95% CI 1.54, 4.58) or an emergency CS (RRR 3.92; 95% CI 2.27, 6.78) than a vaginal delivery. Similarly, overweight or obese women were more likely than healthy weight women to have undergone an elective (RRR 1.91; 95% CI 1.27, 2.87) or an emergency CS (RRR 2.02; 95% CI 1.35, 3.02) than a vaginal delivery. Specialist education of obstetricians and midwives along with financial incentives paid to private hospitals to encourage natural delivery may help in the reduction of unnecessary CS in Iran. In addition, to increase their childbirth knowledge and self-efficacy, pregnant women need to have the opportunity to attend purposefully designed antenatal childbirth preparation classes where they receive evidence-based information on natural childbirth and alternative methods of pain control, as well as the risks and indications for CS.

## 1. Introduction

When used appropriately, Caesarean section (CS) is an important intervention in reducing maternal and perinatal mortality and morbidity. A recent systematic review [1] reported that at the population level CS rates higher than 10% are not associated with reductions in maternal and newborn mortality rates. It is unclear, however, what the optimal rate of CS should be to minimise maternal and infant morbidity outcomes associated with complicated pregnancies [1]. Iran has one of the highest rates of CS in the world, and in 2009, at 47.9%, it was higher than other countries in the Eastern Mediterranean region [2].

The high rates of CS in Iran and other countries suggest that many women undergo a CS for reasons that are not justified on medical grounds. A large proportion of CS in Iran, particularly in private hospitals, are elective CS and performed due to the preferences of the health care provider and/or mother [3,4]. Caesarean sections are not without risk and, as with any surgery, are associated with short- and long-term dangers to the health of mothers and neonates, and may have a negative effect on future pregnancies [5,6].

Advanced maternal age, grand multiparity, obesity, and gestational diabetes are biomedical risk factors associated with an increased likelihood of CS in general [7,8,9,10], while in Iran and other low and middle-income countries, socio-demographic determinants include maternal education, workforce participation, and socio-economic status [11,12,13,14,15,16].

This study reports the incidence of CS and identifies the socio-demographic and biomedical factors associated with delivery by CS of a cohort of healthy, full-term Iranian infants. To our knowledge, this is the first Iranian study to explore the determinants of elective and emergency CS as separate pregnancy outcomes.

## 2. Materials and Methods

### 2.1. Study Design and Setting

This study is a secondary analysis of data collected in the Shiraz, Infant Feeding Survey, a prospective cohort study conducted in Shiraz, Iran, between June 2014 and March 2015. The design of this study has been reported previously [17], but briefly, a cohort of mothers was recruited within 48 h of giving birth in three public and two private maternity hospitals. Participants were followed-up for six months when they attended their local Maternal and Child Health (MCH) clinic for routine postpartum care at 1, 3, 4, and 6 months.

### 2.2. Study Sample

Women were eligible to participate if they were Iranian-born, 18 years of age or older, resided in Shiraz, and had delivered a healthy, full-term (≥37 weeks) infant, weighing 2500 g or more and who had not been admitted to the neonatal intensive care unit (NICU) for 72 h or more. Women were recruited sequentially until the target sample size of 700 was achieved, and recruitment from each hospital was proportionate to the number of infants delivered by the hospital in the preceding year Table 1.

### 2.3. Data Collection

The data used in this cross-sectional analysis were collected from mothers at the time of recruitment via structured face-to-face interviews conducted by a trained research assistant with some data extracted with permission from the participant’s medical record (fasting plasma glucose (FPG) levels and pre-pregnancy weight and height). The questionnaire used by interviewers was based on a questionnaire previously used to study infant feeding practices in a number of Islamic countries [12,18].

### 2.4. Ethical Considerations

The study was approved by the Curtin University Human Research Ethics Committee (HR 31/2014) and the Local Research Ethics Committee of the Shiraz University of Medical Sciences (209/2014). Participation was voluntary and all women provided signed informed consent.

### 2.5. Statistical Analysis

The dependent outcome variable in this analysis was self-reported method of delivery (vaginal, elective CS, emergency CS). No distinction was made as to whether an elective CS was requested by a woman or recommended by her obstetrician. The independent explanatory variables identified in the literature as being associated with CS included maternal age (<25 year, 25–29 year, ≥30 year), educational level (primary to lower secondary, high school, university), pre-pregnancy employment status (employed, unemployed), parity (1 child, 2 children, ≥3 children), pre-pregnancy body mass index (BMI) (normal weight-BMI <25 kg/m^2^, overweight or obese-BMI ≥25 kg/m^2^), a diagnosis of gestational diabetes or pre-pregnancy diabetes (FPG ≥ 92 mg/dL) according to WHO diagnostic criteria [19], attendance at antenatal classes (yes, no), and infant birth weight (2500–2999 g, 2000–3499 g, ≥3500 g).

Data were analysed using IBM SPSS Statistics for Windows, Version 24.0 (IBM Corp. Released 2013. IBM Corp: Armonk, NY, USA). The chi-square test was conducted to examine the bivariate relationship between method of delivery and each of the explanatory variables. As the outcome variable has more than two categorical levels, a multivariable multinomial logistic regression analysis was conducted to investigate the independent association of the explanatory variables and method of delivery. The reference category for the outcome variable was vaginal delivery. All explanatory variables were entered into the initial full model, which was then reduced by using a backward stepwise elimination procedure (cut-off probability for retainment is *p* < 0.05). The model fit was assessed by the likelihood ratio test. The exponentiated results of the multinomial logistic regression are reported as adjusted relative risk ratios (RRRs), and 95% confidence intervals (CIs) were generated. A *p*-value less than 0.05 was considered statistically significant.

## 3. Results

A total of 1571 mothers who delivered in the participating hospitals were invited to participate. Of these, 719 were eligible and 700 (97.4%) agreed to participate. Of the 852 women who failed to meet one or more of the eligibility requirements, 394 resided outside of Shiraz, had delivered an infant that was <37 weeks gestation (*n* = 129), <2500 g (*n* = 280), or were admitted to the NICU for ≥72 h (*n* = 304) (Figure A1).

The majority of mothers had delivered by CS (70.1%) either elective (35.4%) or emergency (34.7%). The overall rate of CS was higher in the private (86.3%) than public hospitals (68.1%) (Table 2). The chi-square tests revealed that method of delivery was associated with maternal age, level of education, pre-pregnancy employment status, pre-pregnancy BMI, and type of hospital where delivery occurred. No association was found between method of delivery and infant birth weight, parity or whether a woman had attended an antenatal class, or had gestational diabetes or pre-pregnancy diabetes.

The multivariable multinomial logistic regression analysis revealed a number of significant independent associations between maternal characteristics and the mode of delivery, with four of the nine explanatory variables remaining in the final model (Likelihood ratio test *p* = 0.001) (Table 3). For instance, women were more likely to deliver by elective CS than vaginally if they were older (≥30 year) compared to younger mothers (<25 year) (RRR 2.22; 95% CI 1.28, 3.84) and had given birth in a private hospital (RRR 3.64; 95% CI 1.79, 7.38), but no such significant associations were found with the likelihood of delivering by emergency CS. Compared to those educated to primary or lower secondary level, university educated women were more likely to have undergone an elective CS (RRR 2.65; 95% CI 1.54, 4.58) or an emergency CS (RRR 3.92; 95% CI 2.27, 6.78) than a vaginal delivery. Similarly, overweight or obese women were more likely than healthy weight women to have undergone an elective CS (RRR 1.91; 95% CI 1.27, 2.87) or an emergency CS (RRR 2.02; 95% CI 1.35, 3.02) than a vaginal delivery. 

## 4. Discussion

Iran has one of the highest rates of CS in the world and inappropriate CS represents a major health system problem [20]. This study revealed that seven out of every ten mothers of healthy, full-term infants delivered by CS, which is comparable to an incidence of 62.2% reported in a contemporaneous study conducted in Shiraz in 2013 [21]. Just over a third of deliveries in this study were purported to be emergency CS, a rate that far exceeds the acceptable rate of 10 to 15% of deliveries recommended by the World Health Organization (WHO) [5], suggesting that many of these may not have been medically indicated. CS rates are highest in Tehran and developed provinces [2], and the increase in CS over the last three decades has been attributed to socioeconomic development and ‘modernisation’ [11]. In this study, approximately half of all CS were elective operations, and a steady increase in elective CS explains, at least in part, the steady rise in overall CS rates [2].

Maternal obesity and gestational diabetes are associated independently with an increased risk of a range of adverse pregnancy outcomes affecting both the mother and the offspring in the short- and long-term [8,9]. Both are associated with an increased incidence of CS [8,9] and macrosomia [9,22]. Therefore, an elective CS may be suggested by an obstetrician or requested by a mother with one or both of these conditions as a means of reducing the maternal fear and pain, and obstetrical difficulties and risks associated with delivering a large birth weight baby vaginally. In this study, there was no independent association between mode of delivery and infant birth weight or a diagnosis of gestational diabetes or pre-pregnancy diabetes; however, compared to healthy weight women, those who were overweight or obese were more likely to undergo either an elective or emergency CS than to deliver vaginally.

Similarly, advanced maternal age is associated with a variety of adverse maternal and perinatal pregnancy outcomes [10], and these risks may inform an obstetrician’s decision to perform an elective CS. In this study, while there was no association with maternal age and emergency CS, compared to younger mothers aged less than 25 years, women aged 25 years and older were more than twice as likely to undergo an elective CS than deliver vaginally. This finding was independent of parity and is consistent with other Iranian studies, which have reported that the rate of CS increases steadily with maternal age [11,13,21].

The rate of CS, was higher in the private hospitals studied where almost nine out of every 10 women delivered by CS compared to just under seven out of every 10 women in the public hospitals. Higher rates of CS have been consistently reported for private hospitals compared to government run hospitals in Iran, and this difference is explained primarily by a higher rate of elective operations in private hospitals [2,11,21,23]. In this study, just over seven out of every 10 women delivered by elective CS in private hospitals compared to three out of every 10 women in public hospitals. While earlier studies have not always differentiated between the determinants of elective and emergency CS, in this study, women delivering in a private hospital were close to four times more likely to have an elective CS than a vaginal delivery, but there was no significant association with emergency CS and type of hospital.

While women were not asked in this study if they had requested a CS, other Iranian studies have reported that women delivering in private hospitals are more likely to request an elective CS than those delivering in public hospitals [23]. For instance, Ghotbi and colleagues reported in a study of public and private hospitals in Tehran that, overall, 20.8% of women requested a CS; however, the rate of maternal request was much higher in private hospitals (42.4%) than in public hospitals (6.1%). Older, employed, and better educated women were the most likely to request a CS [23].

In this study, women educated to primary or lower secondary level were less likely to undergo either an elective or emergency CS than university educated women. This association has been reported consistently in other Iranian studies [11,13,21], and there are a number of probable reasons why less educated woman are more likely to deliver vaginally. Less educated women may come from more traditional backgrounds where vaginal delivery has been the norm in their family and community and, therefore, they may have more positive attitudes towards natural childbirth, or alternatively, they may be fearful of surgery. While a vaginal delivery is free in public hospitals, there is a fee for a CS operation and, therefore, less well-educated women, who are more likely to be of low income, will be less likely for financial reasons to request an elective CS. Conversely, there is a cultural perception of social prestige attached to being able to afford the costs associated with having a CS [4], which helps explain the higher rates among better educated and employed women [23].

Fear of childbirth is universal and a major reason why women elect for a CS [24,25]. Iranian women do not receive health education about the reproductive system and childbirth in school [23,26] and, in general, have poor knowledge about childbirth and various modes of delivery [23]. As a consequence, women’s attitudes and beliefs about childbirth are influenced by normative beliefs [4], and their preference for CS is largely informed by their mother, relatives, and friends who may share negative natural childbirth stories and positive CS stories [26]. Several qualitative studies have investigated the reasons why Iranian women request CS, and these primarily include fear of childbirth (labour pain, injury to mother or infant) and the belief that CS is a safer mode of delivery and that vaginal delivery carries increased risk of complications (vaginal prolapse, urinary incontinence, sexual dysfunction) [4,13,26]. These same views are often held and promoted by obstetricians who describe vaginal deliveries as time consuming and unpredictable, with obstetricians preferring to perform a CS because they perceive them to be safer and that they have more control over the process [3,27].

While CS can be lifesaving under emergency medical conditions there are a number of adverse consequences associated with unnecessary CS. Planned CS have been associated with an increased risk of death and a number of postpartum complications including wound hematoma, hysterectomy, major puerperal infection, anaesthetic complications, and prolonged hospital stay [28]. In addition, a CS is more costly to perform than a vaginal delivery; hence, when performed without medical indication, diverts much needed money from other sectors of the health care system [2]. Finally, CS is associated negatively with a variety of breastfeeding outcomes [18,29]. There was evidence of a negative impact on breastfeeding in this study, and compared to women who delivered vaginally, those who delivered by either elective or emergency CS were less likely to initiate breastfeeding within the first hour and their infant was more likely to receive infant formula during their hospital stay [30], with both these practices being associated with reduced duration of breastfeeding [31].

Research has shown that awareness of the benefits of natural childbirth is associated with a decreased preference for CS [32]. Structured antenatal classes have been shown in a Danish study to increase childbirth self-efficacy, which is associated with lower levels of anxiety and pain [33]. Despite universal access to antenatal care, childbirth education programs are uncommon in the Iranian antenatal care system [34]. A lack of suitable facilities, audio-visual equipment, and education materials are reasons given by health care providers for why Iranian women hardly ever receive group childbirth education [35]. High workloads and staff shortages mean that health care providers have only a few minutes with each woman during antenatal visits to provide advice and answer a woman’s questions [35].

In this study, only one in every eight women attended an antenatal class, and of these, more delivered vaginally (16.3%) than by elective CS (9.3%, *p* = 0.069); however, the numbers were insufficient to detect statistical significance. Nevertheless, a small quasi-experimental study conducted in central Iran provides evidence of the effectiveness of antenatal classes attended by primigravida women [36]. Compared to the control group, there was a significant improvement in the childbirth knowledge and attitudes of the intervention group, and significantly more vaginal deliveries.

The high rate of CS in Iran is recognised as a major health system problem requiring a multi-strategy solution [27]. To this effect, the Iranian Ministry of Health has implemented the Promotion of Natural Childbirth (PNC) program, which reportedly has started to have a modest effect in reducing the rate of CS [20]. In addition to the strategies included in the PNC, further reductions in CS may be seen with the specialist education of health professionals aimed at improving attitudes towards natural delivery and changing their behaviours [3], and if private hospitals were to receive financial incentives for natural delivery [20]. More needs to be done, however, to change the childbirth attitudes and behaviours of mothers [34]. Routine antenatal childbirth preparation classes should be conducted for pregnant women in which the risks and benefits associated with different modes of delivery and methods of pain relief are discussed, and where they can ask questions and express their concerns. If made mandatory, antenatal classes could not only have an immediate effect on CS rates, but a cumulative effect on CS rates potentially would be felt over the long-term as more women have positive natural childbirth experiences and subsequently report these experiences to other women through their family and social networks.

There are several limitations to consider when interpreting the results of this study. The primary limitation is that this study recruited only mothers of healthy, full-term infants. The majority of women deemed ineligible to participate had delivered infants who were either premature, of low birth weight, or had health problems requiring them to be admitted to the NICU for an extended period. Therefore, the incidence of CS in the hospitals studied is likely to be higher than that reported here. Findings are based on self-reported data, and as this was a secondary analysis of data collected as part of infant feeding, information on the reason for the CS and whether it was medically justified and prior history of CS in multiparous women or history of gynaecological diseases was not obtained from the mother, nor were the opinions and medical judgements of the obstetricians available. Despite these limitations, the findings of this study are consistent with those of contemporaneous studies in Shiraz [21] and Tehran [23].

## 5. Conclusions

This study reveals that rates of elective and emergency CS were four to five times higher than that recommended by WHO. CS were more commonly performed at private than public hospitals, and the high and increasing rate of CS suggests that many women undergo a CS for reasons that are not justified on medical grounds. There is an urgent need for a multi-strategy approach to halt and reverse this trend, which includes financial incentives for private hospitals, specialist health professional education, and childbirth education for pregnant women. To increase their childbirth knowledge and self-efficacy, women should have the opportunity to attend group education antenatal classes where they receive evidence-based information on natural childbirth and alternative methods of pain control, as well as the risks of and indications for CS.

## Figures and Tables

**Table 1 ijerph-17-05632-t001:** Number of participants recruited from each hospital.

Hospital	No. Births 2013	%	No. of Participants	%
Public 1	6728	32	220	31
Public 2	5834	28	200	29
Public 3	6073	29	200	29
Private 1	1600	8	60	9
Private 2	684	3	20	3
Total	20,919	100	700	100

**Table 2 ijerph-17-05632-t002:** Characteristics of participants and association with mode of delivery.

Characteristic	Total (*N* = 700)		Vaginal Delivery (*N* = 209)		Elective Caesarean Section (*N* = 248)		Emergency Caesarean Section (*N* = 243)		
	*n*	% ^a^	*n*	% ^b^	*n*	% ^b^	*n*	% ^b^	*p* Value ^c^
Mother’s age (year)									0.001
<25	150	21.4	60	40.0	31	20.7	59	39.3	
25–29	265	37.9	74	27.9	101	28.1	90	34.0	
≥30	285	40.7	75	26.3	116	40.7	94	33.0	
Mother’s education									
Primary to lower secondary	142	20.3	66	46.5	40	28.2	36	25.4	<0.001
High school	287	41.0	88	28.2	91	31.7	108	37.6	
University	271	38.7	55	25.4	117	43.2	99	36.5	
Mother’s pre-pregnancy employment status									
Employed	125	17.9	21	16.8	56	44.8	48	38.4	0.001
Unemployed	575	82.1	188	32.7	192	33.4	195	33.9	
Maternity hospital									
Private	80	11.4	11	13.8	59	73.8	10	12.5	<0.001
Public	620	88.6	198	31.9	189	30.5	233	37.6	
Parity (number of children)									
1	382	54.6	106	27.7	130	34.0	146	38.2	0.153
2	255	36.4	78	30.6	97	38.0	80	31.4	
≥3	63	9.0	25	39.7	21	33.3	17	27.0	
BMI pre-pregnancy ^d^									
Healthy weight	408	58.3	143	35.0	132	32.4	133	32.6	0.002
Overweight or obese	292	41.7	66	22.6	116	39.7	110	37.7	
Gestational diabetes or diabetic ^e^									
Yes	275	39.3	84	30.5	85	30.9	106	38.5	0.100
No	425	60.7	125	29.4	163	38.4	137	32.2	
Attended an antenatal class									
Yes	85	12.1	34	40.0	23	27.1	28	32.9	0.069
No	615	87.9	175	28.5	225	36.6	215	35.0	
Infant birthweight (g)									
2500–2999	224	32.0	55	24.6	92	41.1	77	34.4	0.093
3000–3499	320	45.7	99	30.0	103	32.2	118	36.9	
3500–4000	156	22.3	55	35.3	53	34.0	48	30.8	

^a^ Column percentage; ^b^ Row percentage; ^c^ Chi-square test; ^d^ Healthy weight = BMI < 25 kg/m^2^, Overweight or obese = BMI ≥ 25 kg/m^2^; ^e^ FPG ≥ 92 mg/dL.

**Table 3 ijerph-17-05632-t003:** Maternal characteristics associated with Caesarean (elective or emergency) delivery.

Characteristic	RRR ^a^ of Elective Caesarean Section Compared with Vaginal Delivery			RRR ^a^ of Emergency Caesarean Section Compared with Vaginal Delivery		
	RRR	95% CI	*p* Value	RRR	95% CI	*p* Value
Mother’s age (year)						
<25	1.00			1.00		
25–29	2.01	1.16, 3.49	0.013	1.09	0.66	0.742
≥30	2.22	1.28, 3.84	0.004	1.01	1.78	0.957
Mother’s education						
Primary to lower secondary	1.00			1.00		
High school	1.73	1.04, 2.87	0.034	2.35	1.42, 3.89	0.001
University	2.65	1.54, 4.58	<0.001	3.92	2.27, 6.78	<0.001
Maternity hospital						
Public	1.00			1.00		
Private	3.64	1.79, 7.38	<0.001	0.50	0.20, 1.23	0.133
Maternal pre-pregnancy BMI						
Healthy weight	1.00			1.00		
Overweight/obese	1.91	1.27, 2.87	0.002	2.02	1.35, 3.02	0.001

^a^ Adjusted for mother’s pre-pregnancy employment status, attendance at antenatal classes, parity, gestational diabetes or diabetes in pregnancy, and infant birth weight. RRR adjusted odds ratio, CI confidence interval.

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
