# Peer review of "Incidence and Determinants of Caesarean Section in Shiraz, Iran"

_ijerph, 2020, doi:10.3390/ijerph17165632_

Round 1
Reviewer 1 Report
Incidence and determinants of caesarean section in Shiraz, Iran
Thank you for the opportunity to review the manuscript entitled "Incidence and determinants of caesarean section in Shiraz, Iran". The article corresponds to the subjects and aims of International Journal of Environmental Research and Public Health.
This is an interesting paper, which aim was to study incidence of CS and identifies the sociodemographic and biomedical factors associated with delivery by CS of a cohort of healthy, full-term Iranian infants.
The article possess practical value, the subject is an important one.
The title and the abstract are written in a proper way and the abstract summarizes the content of the article. The article is very interesting, demonstrate by originality. Research results are also practical aspect.
The interpretation of the results is clearly presented and adequately supported by the evidence adduced, but I have a few comments.
Page 3, lines 108-109: The overall rate of CS was higher in the private (86.2%) than public hospitals (68.1%) (Table 2) - This result is not apparent from the data in Table 2.
Table 2. Public - Elective CS: 76.2%, Emergency CS: 95.4%, Private - Elective CS: 23.8%, Emergency CS: 4.1%?
The conclusions are logically valid and justified by the evidence adduced. The quoted literature is selected properly for the subject of the article, and it is current enough. The article is possess scientific and practical value.
After minor adjustments, the manuscript can be published.
Author Response
Comment: Page 3, lines 108-109: The overall rate of CS was higher in the private (86.2%) than public hospitals (68.1%) (Table 2) - This result is not apparent from the data in Table 2.
Table 2. Public - Elective CS: 76.2%, Emergency CS: 95.4%, Private - Elective CS: 23.8%, Emergency CS: 4.1%?
Response: The percentages in Table 2 were all originally column percentages to be consistent with the percentages given for the total sample. We agree with reviewer 1 that the associations between independent variables and type of delivery were not readily discernible. Therefore, we now present these data as row percentages.
Public Elective 30.5%, public emergency 37.6%. Private elective 73.8%, and private emergency 12.5%. Please note that a rounding error was made the overall percentage of CS in public hospitals is 86.3%.
Reviewer 2 Report
This manuscript reports the incidence of CS and identifies the determinants of elective and emergency CS as separate pregnancy outcomes in 700 Iranian women. Those women who are older (≥30 yr) are more likely to deliver by elective CS than vaginally compared to younger mothers (<25 yr) and they prefer to give birth at a private hospital. Women with higher education are more likely to have undergone an elective or an emergency CS than a vaginal delivery. Besides, disease or health condition is also an important factor affecting women's choice of cesarean section or normal delivery. Overweight or obese women are more likely than healthy-weight women to choose an elective or an emergency CS than a vaginal delivery.
There are several suggestions for clarification.
- For those second-time mothers who chose a cesarean section, did they deliver by cesarean section or vaginally when they gave birth for the first time?
- Do women who choose caesarean section have a history of gynecological diseases, for example adenomyosis or endometriosis?
Author Response
Comment: There are several suggestions for clarification.
- For those second-time mothers who chose a cesarean section, did they deliver by cesarean section or vaginally when they gave birth for the first time?
- Do women who choose caesarean section have a history of gynecological diseases, for example adenomyosis or endometriosis?
Response: As this study was a secondary analysis of data collected as part of an infant feeding study we did not collect a detailed reproductive history from women either directly or from their medical notes. We have expanded on this limitation in the discussion and clarified the design of the study in section 2.1
Reviewer 3 Report
Authors must include discussion of the limitations of Chi-Square and logistic regression for this sample size and number of IV's. Example; dichotomizing age (</>25) diminishes any power to detect differences.
While maternal antenatal education is a sound and useful recommendations, I would encourage broader recommendations at various levels of intervention based on this and previous research.
Author Response
Comment: Authors must include discussion of the limitations of Chi-Square and logistic regression for this sample size and number of IV's. Example; dichotomizing age (</>25) diminishes any power to detect differences.
Response: According to the paper "Bujang MA, Sa’at N, Sidik T, et al. Sample size guidelines for logistic regression from observational studies with large population: emphasis on the accuracy between statistics and parameters based on real life clinical data. MJMS. 2018;25(4):122–130", the sufficient sample size for logistic regression in observational studies should be taking a minimum sample size of 500 to be necessary for deriving the statistics that represent the parameters or using n = 100 + 50i where i refers to number of IVs in the final model. Overall we have a sample size of 700, which is larger than 100+50*9, given we have 9 IVs in the initial model and 4 IVs in the final model.
If considering the "event per IV" directly, according to the formula suggested by "Peduzzi P, Concato J, Kemper E, et al. A simulation study of the number of events per variable in logistic regression analysis, J Clin Epidemiol, 1996, vol. 49 (pg. 1373-9)", given the event of interest of the paper (Vaginal delivery =209, Elective Cesarean Section =248, Emergency Cesarean Section 243), we can have a valid logistic regression model containing no more than 19 IVs (i < min (n1, n0)-1 = 19). This study sample only has 9 IVs in the initial model and 4 IVs for the final model, which is much less than the criterion.
In summary, the study has a sufficient sample size and reasonable ratio of event per IV to obtain reliable estimates of regression coefficients in the logistic regression analyses.
We are somewhat confused with the reviewer’s comments related to the age variable, as this variable was not dichotomized and it has three levels. Furthermore, we did detect a significant association with elective CS but not emergency CS. For ease of reporting we have chosen to report maternal age as a categorical IV. However, in a preliminary analysis using age as a continuous variable we found similar results. The association of age and elective CS was RRR 1.058, 95% CI 1.015-1.102, p = 0.008 and the association with emergency CS was RRR 1.025, 95%CI 0.984-1.068, p=0.230
Comment: While maternal antenatal education is a sound and useful recommendations, I would encourage broader recommendations at various levels of intervention based on this and previous research.
Response: We have broadened our recommendations as suggested with the addition of the following sentence.
In addition to the strategies included in the PNC, further reductions in CS may be seen with specialist education of health professionals aimed at improving attitudes towards natural delivery and changing their behaviors [3], and if private hospitals were to receive financial incentives for natural delivery [20]